# A Shading-Guided Generative Implicit Model for Shape-Accurate 3D-Aware Image Synthesis

Xingang Pan[1]    Xudong Xu[2]    Chen Change Loy[3]    Christian Theobalt[1]    Bo Dai[3]

[1]Max Planck Institute for Informatics    [2]The Chinese University of Hong Kong
{xpan,theobalt}@mpi-inf.mpg.de            xx018@ie.cuhk.edu.hk
[3]S-Lab, Nanyang Technological University
{ccloy, bo.dai}@ntu.edu.sg

## Abstract

The advancement of generative radiance fields has pushed the boundary of 3D-aware image synthesis. Motivated by the observation that a 3D object should look realistic from multiple viewpoints, these methods introduce a multi-view constraint as regularization to learn valid 3D radiance fields from 2D images. Despite the progress, they often fall short of capturing accurate 3D shapes due to the *shape-color ambiguity*, limiting their applicability in downstream tasks. In this work, we address this ambiguity by proposing a novel shading-guided generative implicit model that is able to learn a starkly improved shape representation. Our key insight is that an accurate 3D shape should also yield a realistic rendering under different lighting conditions. This *multi-lighting constraint* is realized by modeling illumination explicitly and performing shading with various lighting conditions. Gradients are derived by feeding the synthesized images to a discriminator. To compensate for the additional computational burden of calculating surface normals, we further devise an efficient volume rendering strategy via surface tracking, reducing the training and inference time by 24% and 48%, respectively. Our experiments on multiple datasets show that the proposed approach achieves photorealistic 3D-aware image synthesis while capturing accurate underlying 3D shapes. We demonstrate improved performance of our approach on 3D shape reconstruction against existing methods, and show its applicability on image relighting. Our code will be released at *https://github.com/XingangPan/ShadeGAN*.

## 1  Introduction

Advanced deep generative models, *e.g.*, StyleGAN [1, 2] and BigGAN [3], have achieved great successes in natural image synthesis. While producing impressive results, these 2D representation-based models cannot synthesize novel views of an instance in a 3D-consistent manner. They also fall short of representing an explicit 3D object shape. To overcome such limitations, researchers have proposed new deep generative models that represent 3D scenes as neural radiance fields [4, 5]. Such 3D-aware generative models allow explicit control of viewpoint while preserving 3D consistency during image synthesis. Perhaps a more fascinating merit is that they have shown the great potential of learning 3D shapes in an unsupervised manner from just a collection of unconstrained 2D images. If we could train a 3D-aware generative model that learns accurate 3D object shapes, it would broaden various downstream applications such as 3D shape reconstruction and image relighting.

Existing attempts for 3D-aware image synthesis [4, 5] tend to learn coarse 3D shapes that are inaccurate and noisy, as shown in Fig.1 (a). We found that such inaccuracy arises from an inevitable ambiguity inherent in the training strategy adopted by these methods. In particular, a form of

35th Conference on Neural Information Processing Systems (NeurIPS 2021).

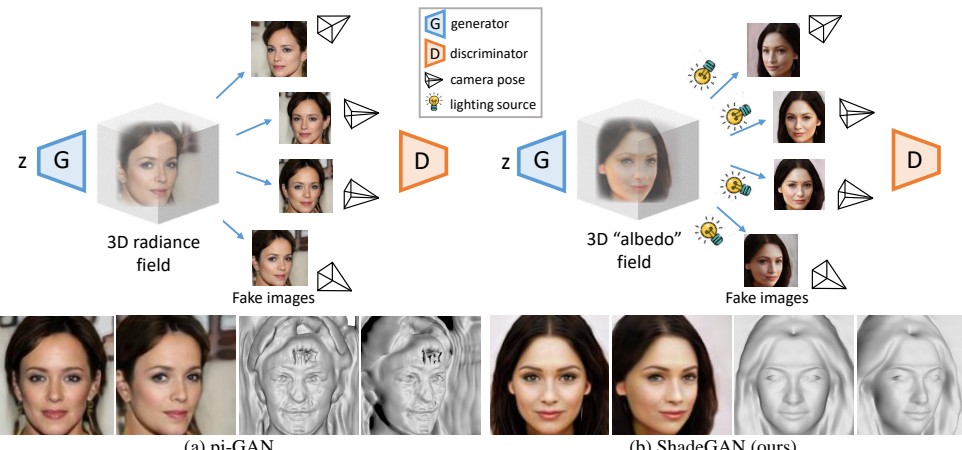

Figure 1: **Motivation**. **(a)** Previous methods like pi-GAN [4] resort to the *"multi-view constraint"*, where the 3D representation is projected to different viewpoints as fake images to the discriminator. The extracted 3D meshes are often inaccurate due to the shape-color ambiguity. **(b)** The proposed approach ShadeGAN further adopts a *"multi-lighting constraint"*, which motivates the 3D representation to look realistic under different lighting conditions. This constraint effectively addresses the ambiguity, giving rise to more natural and precise 3D shapes.

regularization, which we refer to as *"multi-view constraint"*, is used to enforce the 3D representation to look realistic from different viewpoints. The constraint is commonly implemented by first projecting the generator's outputs (*e.g.*, radiance fields [6]) to randomly sampled viewpoints, and then feeding them to a discriminator as fake images for training. While such a constraint enables these models to synthesize images in a 3D-aware manner, it suffers from the *shape-color ambiguity*, *i.e.*, small variations of shape could lead to similar RGB images that look equally plausible to the discriminator, as the color of many objects is locally smooth. Consequently, inaccurate shapes are concealed under this constraint.

In this work, we propose a novel shading-guided generative implicit model (ShadeGAN) to address the aforementioned ambiguity. In particular, ShadeGAN learns more accurate 3D shapes by explicitly modeling shading, *i.e.*, the interaction of illumination and shape. We believe that an accurate 3D shape should look realistic not only from different viewpoints, but also under different lighting conditions, *i.e.*, satisfying the *"multi-lighting constraint"*. This idea shares similar intuition with photometric stereo [7], which shows that accurate surface normal could be recovered from images taken under different lighting conditions. Note that the multi-lighting constraint is feasible as real-world images used for training are often taken under various lighting conditions. To fulfill this constraint, ShadeGAN takes a relightable color field as the intermediate representation, which approximates the *albedo* but does not necessarily satisfy viewpoint independence. The color field is shaded under a randomly sampled lighting condition during rendering. Since image appearance via such a shading process is strongly dependent on surface normals, inaccurate 3D shape representations will be much more clearly revealed than in earlier shading-agnostic generative models. Hence, by satisfying the multi-lighting constraint, ShadeGAN is encouraged to infer more accurate 3D shapes as shown in Fig.1 (b).

The above shading process requires the calculation of the normal direction via back-propagation through the generator, and such calculation needs to be repeated dozens of times for a pixel in volume rendering [4, 5], introducing additional computational overhead. Existing efficient volume rendering techniques [8, 9, 10, 11, 12] mainly target static scenes, and could not be directly applied to generative models due to their dynamic nature. Therefore, to improve the rendering speed of ShadeGAN, we formulate an efficient *surface tracking network* to estimate the rendered object surface conditioned on the latent code. This enables us to save rendering computations by just querying points near the predicted surface, leading to 24% and 48% reduction of training and inference time without affecting the quality of rendered images.

Comprehensive experiments are conducted across multiple datasets to verify the effectiveness of ShadeGAN. The results show that our approach is capable of synthesizing photorealistic images while capturing more accurate underlying 3D shapes than previous generative methods. The learned

distribution of 3D shapes enables various downstream tasks like 3D shape reconstruction, where our approach significantly outperforms other baselines on the BFM dataset [13]. Besides, modeling the shading process enables explicit control over lighting conditions, achieving image relighting effect. Our contributions can be summarized as follows: 1) We address the shape-color ambiguity in existing 3D-aware image synthesis methods with a shading-guided generative model that satisfies the proposed multi-lighting constraint. In this way, ShadeGAN is able to learn more accurate 3D shapes for better image synthesis. 2) We devise an efficient rendering technique via surface tracking, which significantly saves training and inference time for volume rendering-based generative models. 3) We show that ShadeGAN learns to disentangle shading and color that well approximates the albedo, achieving natural relighting effects in image synthesis.

## 2 Related Work

**Neural volume rendering.** Starting from the seminal work of neural radiance fields (NeRF) [6], neural volume rendering has gained much popularity in representing 3D scenes and synthesizing novel views. By integrating coordinate-based neural networks with volume rendering, NeRF performs high-fidelity view synthesis in a 3D consistent manner. Several attempts have been proposed to extend or improve NeRF. For instance, [14, 15, 16] further model illumination, and learn to disentangle reflectance with shading given well-aligned multi-view and multi-lighting images. Besides, many studies accelerate the rendering of static scenes from the perspective of spatial sparsity [8, 9], architectural design [10, 11], or efficient rendering [17, 12]. However, it is not trivial to apply these illumination and acceleration techniques to volume rendering-based generative models [5, 4], as they typically learn from unposed and unpaired images, and represent dynamic scenes that change with respect to the input latent codes.

In this work, we take the first attempt to model illumination in volume rendering-based generative models, which serves as a regularization for accurate 3D shape learning. We further devise an efficient rendering technique for our approach, which shares similar insight with [12], but does not rely on ground truth depth for training and it is not limited to a small viewpoint range.

**Generative 3D-aware image synthesis.** Generative adversarial networks (GANs) [18] are capable of generating photorealistic images of high-resolution, but lack explicit control over camera viewpoint. In order to enable them to synthesis images in a 3D-aware manner, many recent approaches investigate how 3D representations could be incorporated into GANs [19, 20, 21, 22, 23, 24, 25, 26, 27, 5, 4, 28, 29, 30]. While some works directly learn from 3D data [19, 20, 21, 22, 30], in this work we focus on approaches that only have access to unconstrained 2D images, which is a more practical setting. Several attempts [23, 24, 25] adopt 3D voxel features with learned neural rendering. These methods produce realistic 3D-aware synthesis, but the 3D voxels are not interpretable, *i.e.*, they cannot be transferred to 3D shapes. By leveraging differentiable renderer, [26] and [27] learn interpretable 3D voxels and meshes respectively, but [26] suffers from limited visual quality due to low voxel resolution while the learned 3D shapes of [27] exhibit noticeable distortions. The success of NeRF has motivated researchers to use radiance fields as the intermediate 3D representation in GANs [5, 4, 28]. While achieving impressive 3D-aware image synthesis with multi-view consistency, the extracted 3D shapes of these approaches are often imprecise and noisy. Our main goal in this work is to address the inaccurate shape by explicitly modeling illumination in the rendering process. This innovation helps achieve better 3D-aware image synthesis with broader applications.

**Unsupervised 3D shape learning from 2D images.** Our work is also related to unsupervised approaches that learn 3D object shapes from unconstrained, monocular view 2D images. While several approaches use external 3D shape templates or 2D key-points as weak supervisions to facilitate learning [31, 32, 33, 34, 35, 36, 37], in this work we consider the harder setting where only 2D images are available. To tackle this problem, most approaches adopt an "analysis-by-synthesis" paradigm [38, 39, 40]. Specifically, they design photo-geometric autoencoders to infer the 3D shape and viewpoint of each image with a reconstruction loss. While succeed in learning the 3D shapes for some object categories, these approaches typically rely on certain regularization to prevent trivial solutions, like the commonly used symmetry assumption on object shapes [39, 40, 31, 32]. Such assumption tends to produce symmetric results that may overlook the asymmetric aspects of objects. Recently, GAN2Shape [41] shows that it is possible to recover 3D shapes for images generated by 2D GANs. This method, however, requires inefficient instance-specific training, and recovers depth maps instead of full 3D representations.

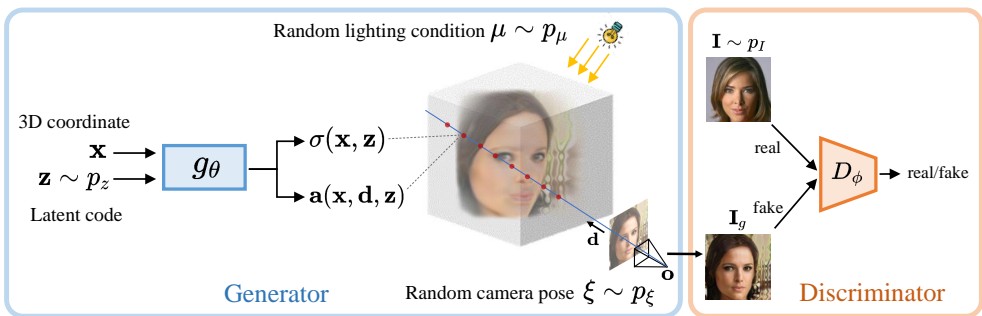

Figure 2: **Method overview**. Our generator $g_\theta$ models a relightable color field conditioned on a latent code $z \sim p_z$. To synthesis an image, it performs volume rendering under a random camera pose $\xi \sim p_\xi$. The rendering process also performs shading with a randomly sampled lighting condition $\mu \sim p_\mu$. The discriminator learns to distinguish the synthesized images with real images from the training dataset, and the whole model is trained with a GAN loss. Although our model is trained from unconstrained 2D images, it allows explicit control over camera pose and lighting condition during inference.

The proposed 3D-aware generative model also serves as a powerful approach for unsupervised 3D shape learning. Compared with aforementioned autoencoder-based methods, our GAN-based approach avoids the need to infer the viewpoint of each image, and does not rely on strong regularizations. In experiments, we demonstrate superior performance over recent state-of-the-art approaches Unsup3d [39] and GAN2Shape [41].

## 3    Methodology

We consider the problem of 3D-aware image synthesis by learning from a collection of unconstrained and unlabeled 2D images. We argue that modeling shading, *i.e.*, the interaction of illumination and shape, in a generative implicit model enables unsupervised learning of more accurate 3D object shapes. In the following, we first provide some preliminaries on neural radiance fields (NeRF) [6], and then introduce our shading-guided generative implicit model.

### 3.1    Preliminaries on Neural Radiance Fields

As a deep implicit model, NeRF [6] uses an MLP network to represent a 3D scene as a radiance field. The MLP $f_\theta : (x, d) \to (\sigma, c)$ takes a 3D coordinate $x \in \mathbb{R}^3$ and a viewing direction $d \in \mathbb{S}^2$ as inputs, and outputs a volume density $\sigma \in \mathbb{R}^+$ and a color $c \in \mathbb{R}^3$. To render an image under a given camera pose, each pixel color $C$ of the image is obtained via volume rendering along its corresponding camera ray $r(t) = o + td$ with near and far bounds $t_n$ and $t_f$ as below:

$$C(r) = \int_{t_n}^{t_f} T(t)\sigma(r(t))c(r(t), d)dt, \text{ where } T(t) = \exp(-\int_{t_n}^{t} \sigma(r(s))ds). \quad (1)$$

In practice, this volume rendering is implemented with a discretized form using stratified and hierarchical sampling. As this rendering process is differentiable, NeRF could be directly optimized via posed images of a static scene. After training, NeRF allows the rendering of images under new camera poses, achieving high-quality novel view synthesis.

### 3.2    Shading-Guided Generative Implicit Model

In this work, we are interested in developing a generative implicit model that explicitly models the shading process for 3D-aware image synthesis. To achieve this, we make two extensions to the MLP network in NeRF. First, similar to most deep generative models, it is further conditioned on a latent code $z$ sampled from a prior distribution $\mathcal{N}(0, I)^d$. Second, instead of directly outputting the color $c$, it outputs a relightable pre-cosine color term $a \in \mathbb{R}^3$, which is conceptually similar to albedo in the way that it could be shaded under a given lighting condition. While albedo is viewpoint-independent, in this work we do not strictly enforce such independence for $a$ in order to account for dataset bias. Thus, our generator $g_\theta : (x, d, z) \to (\sigma, a)$ takes a coordinate $x$, a viewing direction $d$, and a latent

code $z$ as inputs, and outputs a volume density $\sigma$ and a pre-cosine color $a$. Note that here $\sigma$ is independent of $d$, while the dependence of $a$ on $d$ is optional. To obtain the color $C$ of a camera ray $r(t) = o + td$ with near and far bounds $t_n$ and $t_f$, we calculate the final pre-cosine color $A$ via:

$$A(r, z) = \int_{t_n}^{t_f} T(t, z)\sigma(r(t), z)a(r(t), d, z)dt, \text{ where } T(t, z) = \exp(-\int_{t_n}^{t} \sigma(r(s), z)ds). \tag{2}$$

We also calculate the normal direction $n$ with:

$$n(r, z) = \hat{n}(r, z)/\|\hat{n}(r, z)\|_2, \text{ where } \hat{n}(r, z) = -\int_{t_n}^{t_f} T(t, z)\sigma(r(t), z)\nabla_{r(t)}\sigma(r(t), z)dt, \tag{3}$$

where $\nabla_{r(t)}\sigma(r(t), z)$ is the derivative of volume density $\sigma$ with respect to its input coordinate, which naturally captures the local normal direction, and could be calculated via back-propagation. Then the final color $C$ is obtained via Lambertian shading as:

$$C(r, z) = A(r, z)(k_a + k_d\max(0, l \cdot n(r, z))), \tag{4}$$

where $l \in \mathbb{S}^2$ is the lighting direction, $k_a$ and $k_d$ are the ambient and diffuse coefficients. We provide more discussions on this shading formulation at the end of this subsection.

**Camera and Lighting Sampling.** Eq.(2 - 4) describe the process of rendering a pixel color given a camera ray $r(t)$ and a lighting condition $\mu = (l, k_a, k_d)$. Generating a full image $I_g \in \mathbb{R}^{3 \times H \times W}$ requires one to sample a camera pose $\xi$ and a lighting condition $\mu$ in addition to the latent code $z$, i.e., $I_g = G_\theta(z, \xi, \mu)$. In our setting, the camera pose $\xi$ could be described by pitch and yaw angles, and is sampled from a prior Gaussian or uniform distribution $p_\xi$, as also done in previous works [4, 5]. Sampling the camera pose randomly during training would motivate the learned 3D scene to look realistic from different viewpoints. While this *multi-view constraint* is beneficial for learning a valid 3D representation, it is often insufficient to infer the accurate 3D object shape. Thus, in our approach, we further introduce a *multi-lighting constraint* by also randomly sampling a lighting condition $\mu$ from a prior distribution $p_\mu$. In practice, $p_\mu$ could be estimated from the dataset using existing approaches like [39]. We also show in our experiments that a simple and manually tuned prior distribution could also produce reasonable results. As the shading process is sensitive to the normal direction due to the diffuse term $k_d\max(0, l \cdot n(r, z))$ in Eq.(4), this multi-lighting constraint would regularize the model to learn more accurate 3D shapes that produce natural shading, as shown in Fig.1 (b).

**Training.** Our generative model follows the paradigm of GANs [18], where the generator is trained together with a discriminator $D$ with parameters $\phi$ in an adversarial manner. During training, the generator generates fake images $I_g = G_\theta(z, \xi, \mu)$ by sampling the latent code $z$, camera pose $\xi$ and lighting condition $\mu$ from their corresponding prior distributions $p_z$, $p_\xi$, and $p_\mu$. Let $I$ denotes real images sampled from the data distribution $p_I$. We train our model with a non-saturating GAN loss with $R_1$ regularization [42]:

$$\mathcal{L}(\theta, \phi) = \mathbb{E}_{z \sim p_z, \xi \sim p_\xi, \mu \sim p_\mu} \left[ f\left(D_\phi(G_\theta(z, \xi, \mu))\right) \right] + \mathbb{E}_{I \sim p_\mathcal{D}} \left[ f(-D_\phi(I)) + \lambda\|\nabla D_\phi(I)\|^2 \right], \tag{5}$$

where $f(u) = -\log(1 + \exp(-u))$, and $\lambda$ controls the strength of regularization. More implementation details are provided in the supplementary material.

**Discussion.** Note that in Eq.(2 - 4), we perform shading after $A$ and $n$ are obtained via volume rendering. An alternative way is to perform shading at each local spatial point as $c(r(t), d, z) = a(r(t), d, z)(k_a + k_d\max(0, l \cdot n(r(t), z)))$, where $n(r(t), z) = -\nabla_{r(t)}\sigma(r(t), z)/\|\nabla_{r(t)}\sigma(r(t), z)\|_2$ is the local normal. Then we could perform volume rendering using $c(r(t), z)$ to get the final pixel color. In practice, we observe that this formulation obtains suboptimal results. An intuitive reason is that in this formulation, the normal direction is normalized at each local point, neglecting the magnitude of $\nabla_{r(t)}\sigma(r(t), z)$, which tends to be larger near the object surfaces. We provide more analysis in experiments and the supplementary material.

The Lambertian shading we used is an approximation to the real illumination scenario. While serving as a good regularization for improving the learned 3D shape, it could possibly introduce an additional gap between the distribution of generated images and that of real images. To compensate

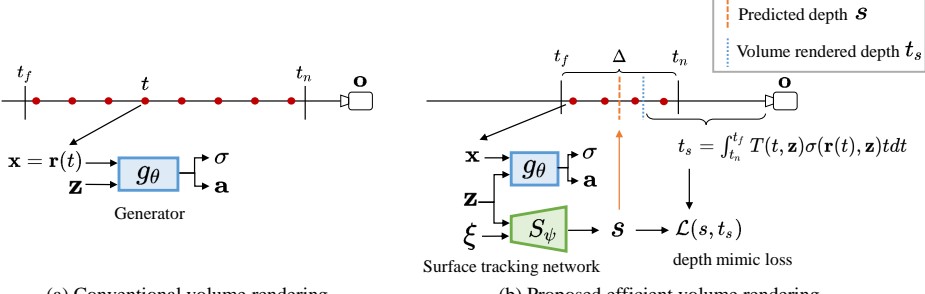

(a) Conventional volume rendering          (b) Proposed efficient volume rendering

Figure 3: (a) Conventional volume rendering samples dozens of points within a predefined near and far bounds $t_n$ and $t_f$. (b) We propose an efficient volume rendering technique via surface tracking. Before rendering, our surface tracking network $S_\psi$ predicts an initial guess of the surface position $s$ conditioned on the latent code $z$ and camera pose $\xi$. Then we sample points near $s$, which requires fewer samples. Finally, we use the volume rendered depth $d$ as the ground truth to train $S_\psi$. During training, $S_\psi$ is able to predict depth $s$ that well approximates the real surface depth $d$.

for such risk, we could optionally let the predicted $a$ be conditioned on the lighting condition, *i.e.*, $a = a(r(t), d, \mu, z)$. Thus, in cases where the lighting condition deviates from the real data distribution, the generator could learn to adjust the value of $a$ and reduce the aforementioned gap. We show the benefit of this design in the experiments.

### 3.3  Efficient Volume Rendering via Surface Tracking

Similar to NeRF, we implement volume rendering with a discretized integral, which typically requires to sample dozens of points along a camera ray, as shown in Fig. 3 (a). In our approach, we also need to perform back-propagation across the generator in Eq.(3) to get the normal direction for each point, which introduces additional computational cost. To achieve more efficient volume rendering, a natural idea is to exploit spatial sparsity. Usually, the weight $T(t, z)\sigma(r(t), z)$ in volume rendering would concentrate on the object surface position during training. Thus, if we know the rough surface position before rendering, we could sample points near the surface to save computation. While for a static scene it is possible to store such spatial sparsity in a sparse voxel grid [8, 9], this technique cannot be directly applied to our generative model, as the 3D scene keeps changing with respect to the input latent code.

To achieve more efficient volume rendering in our generative implicit model, we further propose a *surface tracking network* $S$ that learns to mimic the surface position conditioned on the latent code. In particular, the volume rendering naturally allows the depth estimation of the object surface via:

$$t_s(r, z) = \int_{t_n}^{t_f} T(t, z)\sigma(r(t), z) t \, dt, \tag{6}$$

where $T(t, z)$ is defined the same way as in Eq.(2). Thus, given a camera pose $\xi$ and a latent code $z$, we could render the full depth map $t_s(z, \xi)$. As shown in Fig. 3 (b), we mimic $t_s(z, \xi)$ with the surface tracking network $S_\psi$, which is a light-weighted convolutional neural network that takes $z, \xi$ as inputs and outputs a depth map. The depth mimic loss is:

$$\mathcal{L}(\psi) = \mathbf{E}_{z \sim p_z, \xi \sim p_\xi} \left[ \|S_\psi(z, \xi) - t_s(z, \xi)\|_1 + \text{Prec}(S_\psi(z, \xi), t_s(d(z, \xi))) \right], \tag{7}$$

where Prec is the perceptual loss that motivates $S_\psi$ to better capture edges of the surface.

During training, $S_\psi$ is optimized jointly with the generator and the discriminator. Thus, each time after we sample a latent code $z$ and a camera pose $\xi$, we can get an initial guess of the depth map as $S_\psi(z, \xi)$. Then for a pixel with predicted depth $s$, we could perform volume rendering in Eq.(2,3,6) with near bound $t_n = s - \Delta_i/2$ and far bound $t_f = s + \Delta_i/2$, where $\Delta_i$ is the interval for volume rendering that decreases as the training iteration $i$ grows. Specifically, we start with a large interval $\Delta_{max}$ and decrease to $\Delta_{min}$ with an exponential schedule. As $\Delta_i$ decreases, the number of points used for rendering $m$ also decreases accordingly. Note that the computational cost of our efficient surface tracking network is marginal compared to the generator, as the former only needs a single forward pass to render an image while the latter will be queried for $H \times W \times m$ times. Thus, the reduction of $m$ would significantly accelerate the training and inference speed for ShadeGAN.

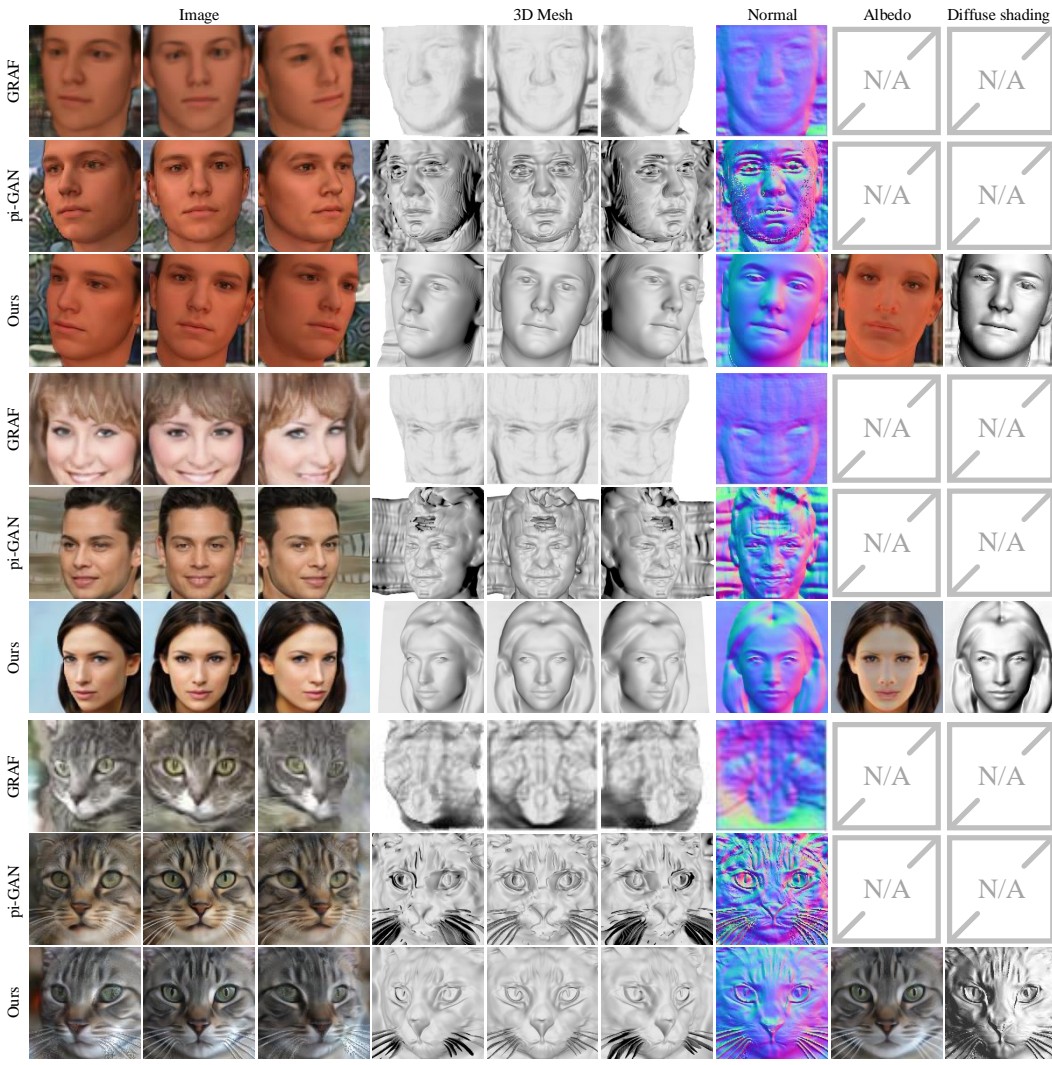

Figure 4: **Qualitative comparison** on BFM (top), CelebA (middle), and Cats (bottom) datasets. "Albedo" refers to the pre-cosine color that approximates albedo. Our approach synthesizes more accurate 3D shapes than pi-GAN and GRAF, and also learns to disentangle shading with albedo.

## 4   Experiments

In this section, we evaluate the proposed ShadeGAN on 3D-aware image synthesis. We also show that ShadeGAN learns much more accurate 3D shapes than previous methods, and in the meantime allows explicit control over lighting conditions. The datasets used include CelebA [43], BFM [13], and Cats [44], all of which contain only unconstrained 2D RGB images.

**Implementation.** In terms of model architectures, we adopt a SIREN-based MLP [45] as the generator and a convolutional neural network as the discriminator following [4]. For the prior distribution of lighting conditions, we use Unsup3d [39] to estimate the lighting conditions of real data and subsequently fit a multivariate Gaussian distribution of $\boldsymbol{\mu} = (\boldsymbol{l}, k_a, k_d)$ as the prior. A hand-crafted prior distribution is also included in the ablation study. In quantitative study, we let the pre-cosine color $\boldsymbol{a}$ be conditioned on the lighting condition $\boldsymbol{\mu}$ as well as the viewing direction $\boldsymbol{d}$ unless otherwise stated. In qualitative study, we observe that removing view conditioning achieves slightly better 3D shapes for CelebA and BFM datasets. Thus, we show results without view conditioning for these two datasets in the main paper, and put those with view conditioning in Fig. 4 of the supplementary material. Other implementation details are also provided in the supplementary.

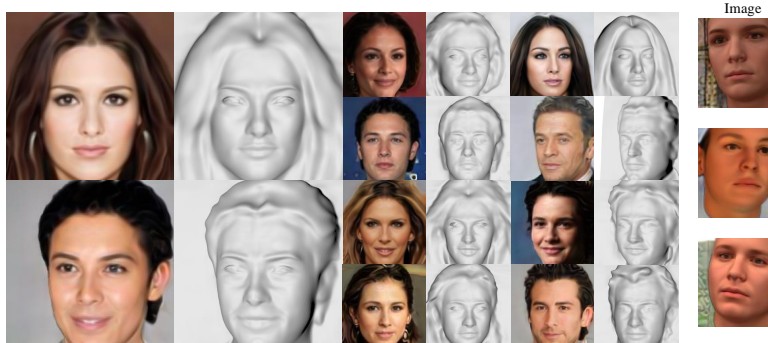

Figure 5: Generated face images and their 3D meshes.

Image | Mesh | Normal | Albedo

(a) ShadeGAN

(b) Local normal

(c) Manual prior

Figure 6: **Qualitative ablation.** See the main text for discussions.

Table 1: **Comparisons on the BFM dataset.** We report FID ($128^2$) for image synthesis, and SIDE ($\times 10^{-2}$) and MAD (deg.) for the accuracy of 3D shapes. '-' indicates not available. Results of pi-GAN and Ours are averaged over 5 runs.

| Method | FID ↓ | SIDE ↓ | MAD ↓ |
|---|---|---|---|
| Supervised | - | 0.410 | 10.78 |
| Unsup3d [39] | - | 0.793 | 16.51 |
| GAN2Shape [41] | - | 0.756 | 14.81 |
| GRAF [5] | 53.4 | 1.857 | 26.60 |
| pi-GAN [4] | $16.7_{\pm 0.2}$ | $0.727_{\pm 0.012}$ | $20.09_{\pm 0.23}$ |
| Ours | $17.7_{\pm 0.2}$ | $0.607_{\pm 0.007}$ | $14.52_{\pm 0.11}$ |

Table 2: **Comparisons on the CelebA and Cats datasets.** The image resolution is $128^2$.

| Dataset | Method | FID ↓ | MAD ↓ |
|---|---|---|---|
| CelebA | GRAF | 43.0 | 30.48 |
| | pi-GAN | 15.7 | 27.22 |
| | Ours | 16.2 | 20.49 |
| Cats | GRAF | 30.3 | 65.47 |
| | pi-GAN | 10.7 | 33.48 |
| | Ours | 10.3 | 25.47 |

**Comparison with baselines.** We compare ShadeGAN with two state-of-the-art generative implicit models, namely GRAF [5] and pi-GAN [4]. Specifically, Fig. 4 includes both synthesized images as well as their corresponding 3D meshes, which are obtained by performing marching cubes on the volume density $\sigma$. While GRAF and pi-GAN could synthesize images with controllable poses, their learned 3D shapes are inaccurate and noisy. In contrast, our approach not only synthesizes photorealistic 3D-consistent images, but also learns much more accurate 3D shapes and surface normals, indicating the effectiveness of the proposed multi-lighting constraint as a regularization. More synthesized images and their corresponding shapes are included in Fig.5. Besides more accurate 3D shapes, ShadeGAN can also learn the albedo and diffuse shading components inherently. As shown in Fig. 4, although not perfect, ShadeGAN has managed to disentangle shading and albedo with satisfying quality, as such disentanglement is a natural solution to the multi-lighting constraint.

The quality of learned 3D shapes is quantitatively evaluated on the BFM dataset. Specifically, we use each of the generative implicit models to generate 50k images and their corresponding depth maps. Image-depth pairs from each model are used as training data to train an additional convolutional neural network (CNN) that learns to predict the depth map of an input image. We then test each trained CNN on the BFM test set and compare its predictions to the ground-truth depth maps as a measurement of the quality of learned 3D shapes. Following [39], we report the scale-invariant depth error (SIDE) and mean angle deviation (MAD) metrics. The results are included in Tab. 1, where ShadeGAN significantly outperforms GRAF and pi-GAN. Besides, ShadeGAN also outperforms other advanced unsupervised 3D shape learning approaches including Unsup3d [39] and GAN2Shape [41], demonstrating its large potential in unsupervised 3D shapes learning. In terms of image quality, Tab. 1 includes the FID [46] scores of images synthesized by different models, where the FID score of ShadeGAN is slightly inferior to pi-GAN in BFM and CelebA. Intuitively, this is caused by the gap between our approximated shading (*i.e.* Lambertian shading) and the real illumination, which can be potentially avoided by adopting more realistic shading models and improving the lighting prior.

In Tab. 2, we also show the quantitative results of different models on CelebA and Cats. To evaluate the learned shape, we use each generative implicit model to generate 2k front-view images and their corresponding depth maps. While these datasets do not have ground truth depth, we report MAD obtained by testing pretrained Unsup3d models [39] on these generated image-depth pairs as

Table 3: **Ablation study on the BFM dataset.**

| No. | Method | FID ↓ | SIDE ↓ | MAD ↓ |
|-----|--------|-------|--------|-------|
| (1) | ShadeGAN | 17.7 | 0.607 | 14.52 |
| (2) | local shading | 30.1 | 0.754 | 18.18 |
| (3) | w/o light | 19.2 | 0.618 | 14.53 |
| (4) | w/o view | 18.6 | 0.622 | 14.88 |
| (5) | manual prior | 20.2 | 0.643 | 15.38 |
| (6) | +efficient | 18.2 | 0.673 | 14.72 |

Table 4: **Training and inference time cost on CelebA.** The efficient volume rendering significantly improves training and inference speed.

| Method | Train (h) | Inference (s) | FID |
|--------|-----------|---------------|-----|
| ShadeGAN | 92.3 | 0.343 | 16.4 |
| +efficient | 70.2 | 0.179 | 16.2 |
| pi-GAN | 56.8 | 0.204 | 15.7 |
| +efficient | 46.9 | 0.114 | 15.9 |

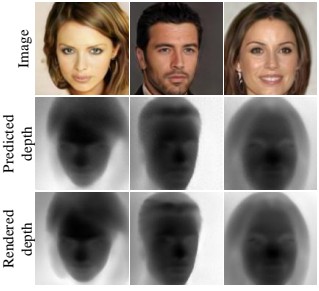

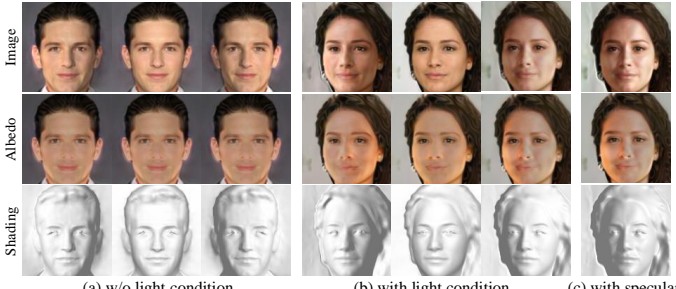

(a) w/o light condition      (b) with light condition      (c) with specular

Figure 7: Visualization of depths predicted by our depth tracking network and those calculated via volume rendering.

Figure 8: **Illumination-aware image synthesis.** ShadeGAN allows explicit control over the lighting. The pre-cosine color (albedo) is independent of lighting in (a) and is conditioned on lighting in (b). We show results of adding a specular term in (c).

a reference. As we can observe, results on CelebA and Cat are consistent with those on the BFM dataset.

**Ablation studies.** We further study the effects of several design choices in ShadeGAN. First, we perform local points-specific shading as mentioned in the discussion of Sec. 3.2. As Tab. 3 No.(2) and Fig. 6 (b) show, the results of such a local shading strategy are notably worse than the original one, which indicates that taking the magnitude of $\nabla_{x}\sigma$ into account is beneficial. Besides, the results of Tab. 3 No.(3) and No.(4) imply that removing $a$'s dependence on the lighting $\mu$ or the viewpoint $d$ could lead to a slight performance drop. The results of using a simple manually tuned lighting prior are provided in Tab. 3 No.(5) and Fig. 6 (c), which are only moderately worse than the results of using a data-driven prior, and the generated shapes are still significantly better than the ones produced by existing approaches.

To verify the effectiveness of the proposed efficient volume rendering technique, we include its effects on image quality and training/inference time in Tab. 3 No.(6) and Tab. 4. It is observed that the efficient volume rendering has marginal effects on the performance, but significantly reduces the training and inference time by 24% and 48% for ShadeGAN. Moreover, in Fig. 7 we visualize the depth maps predicted by our surface tracking network and those obtained via volume rendering. It is shown that under varying identities and camera poses, the surface tracking network could consistently predict depth values that are quite close to the real surface positions, so that we can sample points near the predicted surface for rendering without sacrificing image quality.

**Illumination-aware image synthesis.** As ShadeGAN models the shading process, it by design allows explicit control over the lighting condition. We provide such illumination-aware image synthesis results in Fig.8, where ShadeGAN generates promising images under different lighting directions. We also show that in cases where the predicted $a$ is conditioned on the lighting condition $\mu$, $a$ would slightly change *w.r.t.* the lighting condition, *e.g.*, it would be brighter in areas having a overly dim shading in order to make the final image more natural. Besides, we could optionally add a specular term $k_s \max(0, h \cdot n)^p$ in Eq. 4 (*i.e.*, Blinn-Phong shading [47], where $h$ is the bisector of the angle between the viewpoint and the lighting direction) to create specular highlight effects, as shown in Fig.8 (c).

**GAN inversion.** ShadeGAN could also be used to reconstruct a given target image by performing GAN inversion. As shown in Fig. 9 such inversion allows us to obtain several factors of the image, including the 3D shape, surface normal, approximated albedo, and shading. Besides, we can further perform view synthesis and relighting by changing the viewpoint and lighting condition. The implementation of GAN inversion is provided in the supplementary material.

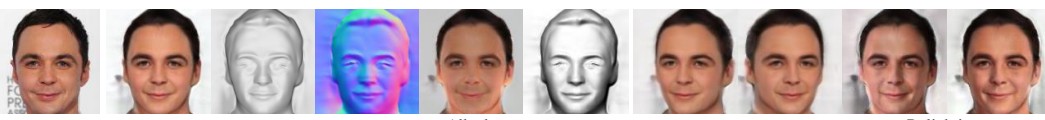

| Real image | Reconstruction | 3D Mesh | Normal | Albedo | Diffuse shading | View synthesis | Relighting |

Figure 9: **GAN inversion for real image editing**.

**Discussions.** As the Lambertian shading we used is an approximation to the real illumination, the albedo learned by ShadeGAN is not perfectly disentangled. Our approach does not consider the spatially-varying material properties of objects as well. In the future, we intend to incorporate more sophisticated shading models to learn better disentangled generative reflectance fields.

## 5 Conclusion

In this work, we present ShadeGAN, a new generative implicit model for shape-accurate 3D-aware image synthesis. We have shown that the multi-lighting constraint, achieved in ShadeGAN by explicit illumination modeling, significantly helps learning accurate 3D shapes from 2D images. ShadeGAN also allows us to control the lighting condition during image synthesis, achieving natural image relighting effects. To reduce the computational cost, we have further devised a light-weighted surface tracking network, which enables an efficient volume rendering technique for generative implicit models, achieving significant acceleration on both training and inference speed. A generative model with shape-accurate 3D representation could broaden its applications in vision and graphics, and our work has taken a solid step towards this goal.

**Acknowledgment.** We would like to thank Eric R. Chan for sharing the codebase of pi-GAN. This study is supported under the ERC Consolidator Grant 4DRepLy (770784). This study is also supported under the RIE2020 Industry Alignment Fund – Industry Collaboration Projects (IAF-ICP) Funding Initiative, as well as cash and in-kind contribution from the industry partner(s).

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
