# OpenReview forum: "A Shading-Guided Generative Implicit Model for Shape-Accurate 3D-Aware Image Synthesis"
_NeurIPS.cc/2021/Conference — NeurIPS 2021 Poster_

### Official Review · Reviewer_YNf7 · 2021-07-06

**Rating:** 4
**Confidence:** 4

**Summary:**

The paper proposes a novel generative model for neural radiance field, where the key idea is to synthetically add shading effects based on random lighting configurations. Shading will make images unrealistic when the geometry is corrupted. Therefore, the system gets trained to generate accurate shapes in its implicit radiance field representation. The system makes convincing results both qualitatively and quantitatively.

**Ethical Concerns:**

As described above, I wish to see some texts explaining potential malicious use of the presented technique (e.g., deep fakes).

**Limitations And Societal Impact:**

I did not find texts addressing the limitations and societal impact. The check-seat has a comment saying that "see the supplementary document" on this point, but I do not see relevanat texts.  Deep fakes are relevant societal impac.

**Main Review:**

The idea of the paper is very simple and intuitive (adding lighting effects and utilizing discriminator to discover accurate geometry). I am not familiar with the complete literature but if no papers have shown this idea before, this idea could be an interesting one for the computer vision community but not for the machine learning community. I am curious if other reviewers spot any existing works with similar ideas. Given extensive GAN literature, there may exist similar ideas somewhere.

Empirical evaluations (both qualitative and quantiative) look convincing, where experimental results also support the claims qualitively and quantitatively. However, they are shown only for human faces.

I am stuck on the overall recommendation but am leaning towards negative. The first reason is the merit/impact to the machine learning community. The strength of this paper is the idea which is for the computer vision community. The technical contribution is weak, as there is not much to learn from the paper. The second reason is that results are only presented for faces. This is very critical. We already have very good approaches that turn single image into 3D face models. This paper will not bring any impacts. The idea of this paper will really shine if this idea works on more variety of object categories. With those reasons, I will vote for rejection.

**Time Spent Reviewing:**

4

---

> ### Author Response · Authors · 2021-08-10
> **Reply to Reviewer YNf7**
>
> Thank you for your review. Below we address the concerns.
>
> **Q1: The merit/impact is for the CV community rather than the ML community.**
>
> **A1**: We believe that our submission fits well with the topic requirements of the NeurIPS conference. Here are the reasons:
> 1) According to the official website of NeurIPS (https://neurips.cc/Conferences/2021/CallForPapers), NeurIPS 2021 is an **"interdisciplinary conference"** that is open to research on computer vision and unsupervised learning. There is no requirement that the merit/impact have to strictly fall in the ML community rather than other communities. On the contrary, the website mentions that "we welcome interdisciplinary submissions that do not fit neatly into existing categories". Some previous works with similar topics are also published on NeurIPS like GRAF [5] and NSVF [7].
> 2) Besides, our submission is not only a computer vision application, but also a state-of-the-art approach for unsupervised learning 3D shapes from unpaired 2D image collections, which also contributes to the unsupervised learning field under ML.
>
> **Q2: There is not much to learn from the paper.**
>
> **A2**: We would like to highlight our contributions here and we hope this could help the reviewer to better understand the merit of this work.
> 1) This work solves an important problem. Recently, there is a trend going from 2D to 3D-aware visual synthesis. Pi-GAN [4], as a state-of-the-art 3D-aware synthesis approach, is known to suffer from inaccurate shape. This work addresses this problem and learns much more accurate 3D shapes than existing GAN approaches and unsupervised learning approaches.
> 2) Our first technical contribution is the "multi-lighting constraint", which is novel in the literature of GAN. The results show that modeling shading and learning from illumination cues in the data is essential for inferring accurate 3D shape. We believe that this idea is inspiring for future work, as also mentioned by Reviewer 8VoF.
> 3) Our second technical contribution is an efficient volume rendering technique that effectively avoids redundant point sampling. This is the first acceleration technique that works with generative radiance field models, which makes the calculation of surface normal tractable.
>
> Besides, we have also studied several model design choices in experiments (Table 3 and Fig. 6), which provides useful experience and guidance for future works.
>
> While the reviewer finds "there is not much to learn from the paper", we hope that the reviewer could provide some explanation or evidence on how the proposed techniques are valueless if possible so that we could better improve this work.
>
> **Q3: Results are only presented for human faces.**
>
> **A3**: First, we would like to clarify that our empirical evaluations are not limited to human faces, but also include a **cats** dataset [43], as shown in **Fig. 4** and **Table 2**. The results already verify that our approach is not designed specifically for human faces and is also applicable to other object categories.
>
> Second, we have tested our approach on more object categories including cars and many wild animals including tiger, lion, cheetah, and wolf, corresponding to the CARLA [1] and AFHQ [2] datasets. In these categories, our approach still learns better 3D shapes than pi-GAN in a similar way as for human faces and cats. The quantitative results on the AFHQ dataset are shown below:
>
> | Method | FID $\downarrow$ | SIDE $\downarrow$ | MAD $\downarrow$  |
> |:------:|------|-------|-------|
> | pi-GAN | 32.2 | 2.208 | 39.17 |
> |  Ours  | 32.6 | 1.824 | 32.92 |
>
> Our approach is not limited to human faces because it by design only requires object-centric image data with lighting variations, and does not involve any specific design for human faces. We will add results on more object categories in the revision.
>
> Third, we believe that our approach still has merits given existing 3D face reconstruction approaches. This is because they are different academic problems with quite different settings, i.e., the 3D shapes in most 3D face reconstruction methods are **supervised** while that in our approach is **unsupervised**. 3D face reconstruction usually requires 3D morphable models (3DMM) [3] that are learned from ground truth 3D face scans collected with professional scan equipment. In contrast, our work studies a more challenging unsupervised learning setting where the 3D shape is learned merely from raw unpaired RGB images. We believe that the value of an unsupervised approach shouldn’t be fully denied by a supervised approach with similar performance.
>
> **Q4: Limitations and societal impact are not found.**
>
> **A4**: We have discussed the limitation of our approach in the “Discussions” paragraph in line 300-303. We have also discussed the broader impacts of this work in Sec. 3 of the supplementary material, which mentions that this work could be used to edit human portraits and thus should strictly respect personality rights and privacy regulations (line 93-95). We will provide discussions on deep fakes and other inappropriate usages of the presented technique in the revision.
>
> With respect to the concern of similar ideas, we did not find similar ideas in the GAN literature according to our survey. Other reviewers also acknowledge the novelty of the idea. We would like to kindly encourage the reviewer to discuss with other reviewers about the novelty and other concerns if possible. Thanks.
>
> [1] Dosovitskiy, Alexey, et al. "CARLA: An open urban driving simulator." Conference on robot learning. PMLR, 2017.
> [2] Choi, Yunjey, et al. "Stargan v2: Diverse image synthesis for multiple domains." CVPR. 2020.
> [3] Blanz, Volker, and Thomas Vetter. "A morphable model for the synthesis of 3D faces." Proceedings of the 26th annual conference on Computer graphics and interactive techniques. 1999.

---

> > ### Comment · Reviewer_YNf7 · 2021-08-18
> > **Switchin from leaning towards reject to learning towards accept**
> >
> > I now lean towards acceptance instead of towards rejection. I had 2 issues.
> >
> > 1. I was concerned on the impact to the ML community but the other reivewers do not have any issues. I do not consider this as an issue any more.
> >
> > 2. It was only shown for human faces (a few cats examples). Authors provides new experimental results on other categories with a table. But this validation is very shallow without any details, and my issue still remains.
> >
> > Overall, one of the issue was gone. I like the overall idea which is simple and intuitive. I now lean towards acceptance.

---

> > > ### Author Response · Authors · 2021-08-19
> > > **Thank you for your feedback**
> > >
> > > Thank you for your feedback. We are glad that we partially address your concerns. We will add more qualitative studies on other categories in the revision.

---

> > > ### Author Response · Authors · 2021-08-29
> > > **A reminder**
> > >
> > > We thank the reviewer again for leaning towards acceptance. We would like to remind the reviewer to kindly update final rating before the end of the discussion period. Thanks.

---

### Official Review · Reviewer_8VoF · 2021-07-16

**Rating:** 7
**Confidence:** 4

**Summary:**

This paper proposes a new method to train generative radiance fields that can achieve more accurate 3D shape reconstruction. The key idea is that instead of directly synthesizing the images from different viewpoints through volume ray tracing, which may suffer from color-shape ambiguity, it synthesizes albedo and normal and then renders the image with different lighting conditions. This new method requires the synthesized albedo and normal to be able to render realistic images under various lighting conditions, providing an extra regularization for shape reconstruction. To further improve the rendering speed, it uses an extra 2D CNN to predict a depth map from latent code and camera poses so that it only needs to sample near the surface in volume ray tracing. Experiments show that the proposed method greatly improves shape reconstruction with similar image synthesis quality compared to state-of-the-arts.

**Ethical Concerns:**

I cannot see any ethical concerns.

**Limitations And Societal Impact:**

Authors mention in the discussion paragraph that one limitation is that the current method does not handle spatially varying material. I cannot see any other obvious limitations that need to be mentioned in the paper. I cannot see any potential negative societal impact either.

**Main Review:**

I will list the strengths of the paper and some minor suggestions that I believe may help improve the paper. My current evaluation of the paper is very positive. Authors may prioritize answering other reviewers' questions in the rebuttal.

Strengths:

1. The major contribution of the paper is clear. It proposes the multi-lighting constraint to help regularize the 3D shape reconstruction when training generative radiance fields, which is intuitively reasonable practically effective. In my opinion, this is an interesting idea and may inspire future research in the related fields.

2.  The paper is clearly written, well-organized, and easy to follow. The citations is complete and appropriate.

3. The experiments are comprehensive and convincing. It clearly supports the major contribution of the paper that the depth prediction quality can be improved by the multi-lighting constraint. Moreover, ablation studies also demonstrate that all minor design improvements such as depth prediction to reduce sampling number and lighting parameters as inputs can further improve the image synthesis quality.

4. Authors clearly summarize the limitations of the current method, which may help researchers to build new frameworks based on current results.

Minor suggestions:

1. When first reading the paper, I was confused by how GAN inversion is done in practice. I later found the details in the supplementary but authors can consider adding a reminder pointing to the supplementary in the main paper.

2. When discussing the intuition of adding the multi-lighting constraint, authors can consider connecting it to photometric stereo, which can achieve accurate normal reconstruction from images taken under different lighting conditions. This may make the argument more convincing and easy to understand.

3. One way to improve the rendering quality is that instead of using sampled lighting conditions, authors can use captured real HDR environment maps. This may minimize the domain gap and render more realistic results. It may be even more useful if authors hope to reconstruct specular highlights in the future.

4. One minor thing to check is if both the lighting and albedo are in the sRGB space or linear RGB space. If they are not in the color space, authors may consider using gamma correction to align them.

**Time Spent Reviewing:**

4

---

> ### Author Response · Authors · 2021-08-10
> **Reply to Reviewer 8VoF**
>
> Thank you for your positive review and valuable suggestions.
>
> **Q1: Adding a reminder pointing to supplementary for GAN inversion.**
>
> **A1**: Thanks for pointing it out. We will add a reminder in the main paper.
>
> **Q2: Connecting to photometric stereo in discussion.**
>
> **A2**: Good idea. We will add this discussion.
>
> **Q3: Use captured real HDR environment maps.**
>
> **A3**: This is a good extension. In this work, we focus on the setting where only raw RGB images are involved. Real HDR environment maps could further improve the rendering quality but would require additional equipment to capture the data. This could be left as future work.
>
> **Q4: Check the color space of lighting and albedo.**
>
> **A4**: Thanks for the suggestion. The albedo is in the sRGB space while the Lambertian shading is linear. This follows [38] and works well in practice. We will try aligning them with gamma correction.

---

> > ### Comment · Reviewer_8VoF · 2021-08-18
> > **No more questions**
> >
> > Thanks authors for the feedback! I have no more questions. My only minor suggestion to align the color space in the final version.

---

> > > ### Author Response · Authors · 2021-08-25
> > > **Thanks**
> > >
> > > Thanks for your comments. We will align the color space in the revision.

---

### Official Review · Reviewer_mEVj · 2021-07-16

**Rating:** 7
**Confidence:** 4

**Summary:**

This paper aims to improve the density distribution of generative radiance field models by generating relitable reflectance fields and using different lightings to shade them during training. The core idea is that artifacts will arise when the generated density distribution ("shape") is unnatural and is getting shaded by different lightings, which should be detected and resolved by a discriminator. Experiments verify that this shading regularization can yield naturally looking shapes induced by the generated radiance fields.

**Limitations And Societal Impact:**

The authors have discussed the mentioned issues to some extent, but not fully addressed them. Please see above comments.

**Main Review:**

---------------UPDATED AFTER BURETTAL

I feel the reponses have addressed my concerns. I recommend acceptance. My suggestion is to add the explanation for view-dependence (to address dataset biases, in addition to non-Lambertian shading) to the main paper.


---------------ORIGINAL REVIEW

Strengths:
+ Simple and interesting idea to improve generative radiance fields.
+ Novel in the context of generative radiance fields.
+ Clearly written.
+ Straightforward implementation with good qualitative results that achieve the goal.

Weakness:
- My major concern is on the unconstrained nature of the "albedo" fields. It is dependent on direction and lighting, making it unconstrained, and possibly unstable, to regularize the density distribution. The "color-shape" ambiguity is still not addressed. In the proposed formulation, the ambiguity can still be baked into the dependency on direction and lighting, i.e., instead of optimizing a good shape, the model might choose to optimize a sophisticated but working A(r,z,d,mu) function. For example, if a generated face shape has no nose, to have a plausible appearance that is multi-view consistent, the model can either (1) generate a nose in the shape and generate a constant albedo in A(z), or (2) keep the "no nose face" and hack the A(r,z,d,mu) function by giving high albedo to left nose when light source is on the left, and low albedo when light source is on the right. This is not seen in the shown images but this might happen to some extent, or might happen to some random seed. Showing results averaged over multiple runs could alleviate this concern and I might increase my rating.

- I understand that the flexible A() function is used also to compensate the simplified lighting and reflectance model. But as for the dependence of view angle, why not use a simple specular part in the BRDF? Just like the test in Figure 8c.

**Time Spent Reviewing:**

3h

---

> ### Author Response · Authors · 2021-08-10
> **Reply to Reviewer mEVj**
>
> Thank you for your constructive comments. Below we address the concerns.
>
> **Q1: Ambiguity may still be baked into the model.**
>
> **A1**: In practice, the inductive bias of our approach tends to avoid the ambiguity of viewpoint and lighting dependence. This phenomenon is also explained in NeRF++ [1]. On one hand, generating a nose in the shape is an easier solution than learning to cheat in the “no nose” case to the model. On the other hand, the conditioning of viewpoint and lighting is introduced in the very last part of the MLP and thus only subtly affects the results. As shown in Table 3, adding viewpoint or lighting conditioning slightly improves the FID score without affecting the 3D shape.
>
> We provide results averaged over 5 runs on the BFM dataset below:
>
> |  Method  | FID $\downarrow$       | SIDE $\downarrow$         | MAD $\downarrow$         |
> |:--------:|------------|---------------|--------------|
> |  pi-GAN  | 17.3 ($\pm$0.2) | 0.725 ($\pm$0.006) | 20.14 ($\pm$0.10) |
> | Ours | 19.2 ($\pm$0.2) | 0.608 ($\pm$0.005) | 14.54 ($\pm$0.08) |
>
> It is observed that the results are stable across multiple runs, verifying our discussions above.
>
> [1] Zhang, Kai, et al. "Nerf++: Analyzing and improving neural radiance fields." arXiv preprint arXiv:2010.07492 (2020).
>
> **Q2: For the dependence of view angle, why not use a specular term?**
>
> **A2**: The dependence of view angle is not just for non-Lambertian effects, but mainly for handling dataset bias, e.g., the eyes of human faces often direct to the camera. So in practice it is still beneficial to add this dependence, which improves the FID score. Adding a specular term in training requires to handle the spatially-varying material properties for realistic synthesis, which is out of the scope of this work.

---

> > ### Comment · Reviewer_mEVj · 2021-08-20
> > **Thanks for response**
> >
> > Thanks for answering my questions! I recommend acceptance and have raised my score.

---

### Official Review · Reviewer_TVTi · 2021-07-18

**Rating:** 6
**Confidence:** 4

**Summary:**


This paper proposes a GAN capable of generating a relightable radiance field when trained on an unlabeled dataset of face images. The relightable radiance field generator is conditioned on the 3D location and a latent code and outputs a volume density field (à la NeRF) and an "albedo" field (I strongly advise against the use of this word; see below) that allows explicit control of the light location. The training is conducted in the GAN style, where the training signal comes from only a discriminator loss. The intuition is that the generator is better off generating a meaningful 3D volume that, when rendered from a random camera view, generates a photorealistic face image that falls into the training data distribution, than generating some intricately-designed volume that gives faces of different identities when viewed from different angles.

The authors demonstrate that the model is able to learn 3D shapes in an unsupervised fashion from just 2D images. The work is mainly compared against pi-GAN, which is shown to suffer from what the authors call "color-shape ambiguities." For efficient rendering, the authors also propose an auxiliary network that predicts the surface location given the latent code and a viewing direction, such that the network can sample around the predicted surface to avoid expensive sampling of unoccupied space.

**Limitations And Societal Impact:**

Yes.

**Main Review:**

The paper explores a cute idea: by additionally modeling lighting, one learns better 3D shapes in an unsupervised way from unstructured 2D images. This gain seems free by exploiting the lighting variation that already exists in the datasets.

The model design is sensible and simple (in a positive sense). The observation that bad 3D shapes do not affect view synthesis in approaches that do not model lighting like pi-GAN is interesting. The shape results look convincing. The insight that the "local shading" alternative underperforms the adopted shading scene is appreciated. The paper is well-written and easy to follow.

Now the drawbacks:

I strongly advise against calling the generated radiance field the "albedo field." Although the authors clarify in the text that this "albedo" depends on the viewing direction and lighting and hence is not the conventional albedo, this makes the paper really hard to read since any vision and graphics person is familiar with the concept that albedo does not depend on viewing directions or lighting. I had to keep reminding myself of the fact that this "albedo" is special. If the properties contradict the two fundamental properties of real albedo, why use it? Maybe something like "pre-cosine radiance" or "unmodulated/demodulated radiance"?

Related to this albedo complaint, the authors applied Lambertian shading to the "albedo" to allow explicit control of lighting. A simpler alternative is directly conditioning the albedo field generation on the light direction also, eliminating the need for a post-hoc Lambertian shading process. While I can see this alternative may not provide direct control over lighting, it warrants an ablation study. If this simpler alternative ends up working better, the authors can consider naming the field something like "light transport field"? See https://arxiv.org/pdf/2008.03806.pdf and https://arxiv.org/pdf/1911.11530.pdf. The current post-hoc Lambertian shading is a bit awkward, without a solid reason for its existence.

In terms of result quality, I'm concerned on the diversity of what the GAN is able to generate. The results shown seem to be suffering from mode collapse. This concern can be easily addressed by showing a video where we vary the latent code and in the meanwhile change the viewpoint and light direction. Then one would be able to judge if the generated results are diverse or not. Not asking for new experiments, but would love to see this in the revision or a future version of this paper.

**Time Spent Reviewing:**

2

---

> ### Author Response · Authors · 2021-08-10
> **Reply to Reviewer TVTi**
>
> Thank you for your constructive comments. Below we address the concerns.
>
> **Q1: The term "albedo field" should be replaced.**
>
> **A1**: Thanks for pointing it out. We agree that “albedo” could be a little bit confusing, and we will replace it with “pre-cosine radiance” in the revision.
> We wish to point out that the dependence of albedo on viewpoint and lighting is not a crucial design in our approach. As shown in Table 3, even without such dependence, our approach still produces much more accurate shapes than pi-GAN, thanks to the proposed multi-lighting constraint. Allowing such dependence achieves a slightly better FID score as it could better handle the dataset bias, e.g., the eyes of human faces often direct to the camera.
>
> **Q2: Lambertian shading may not be necessary.**
>
> **A2**: In our approach, explicitly modeling the shading process is necessary to regularize the underlying 3D shapes. In the alternative where the radiance field is conditioned on the light direction, the shading process is implicitly fused into the network. In this case, the model could generate natural images without having to infer accurate 3D shapes. We have tried this alternative, and it performs similarly to pi-GAN, i.e., its shape is not accurate and lighting could not be explicitly controlled.
>
> The two "light transport field" papers have very different settings from ours. For each instance, they have multiple training images of various viewpoints and lightings to make learning the light transport field possible. In contrast, we have only one image with unknown lighting. Thus, it is infeasible to directly adopt their "light transport field" design in our setting.
>
> As a preliminary attempt, this work shows that Lambertian shading is a simple but effective way to regularize the 3D shape. In experiments, we have also demonstrated the effects of using Blinn-Phong shading in Fig. 8, which could model specular highlight effects. Exploring more advanced shading models is left as future work.
>
> **Q3: Diversity of generated results.**
>
> **A3**: We recommend reviewers to take a look at our video in the supplementary material, which already shows the effects of varying the latent code and in the meanwhile changing the viewpoint. The continuous changing between several different face identities and the examples in Fig. 5 well verifies the image diversity. We are pleased to add more results showing the image diversity in the revision.

---

> > ### Comment · Reviewer_TVTi · 2021-08-22
> > **One More Question**
> >
> > I thank the authors for providing responses to my questions.
> >
> > I'm happy with A1 and A3 but please include results where you vary identity, viewpoint, and lighting all at once in the revision (the current results show only identity change together with viewpoint change).
> >
> > Re: A2
> >
> > Am I right that there is currently no ablation study that proves "explicitly modeling the shading process is necessary to regularize the underlying 3D shape"? The authors mentioned that they've done this experiment. In this case, would the authors share in detail how that experiment was done and what the results look like?

---

> > > ### Author Response · Authors · 2021-08-25
> > > **Reply to Reviewer TVTi**
> > >
> > > Thank you for your feedback.
> > >
> > > We will include results where identity, viewpoint, and lighting are changed all at once in the revision. Thanks for the suggestion.
> > >
> > > As for A2, what we did is that we let the radiance field of pi-GAN be conditioned on not only the viewpoint but also a lighting direction sampled from a prior distribution. The only difference between this pi-GAN-light revision and ShadeGAN is that the former does not explicitly model shading, i.e., Eq(4) in the paper. We observe that the qualitative results of pi-GAN-light are indistinguishable from pi-GAN. The quantitative results below show that explicitly modeling shading significantly improves 3D shape quality.
> > >
> > > |      Method     | FID$\downarrow$  | SIDE$\downarrow$  | MAD$\downarrow$   |
> > > |:---------------:|------|-------|-------|
> > > |   pi-GAN  | 15.7 | 1.453 | 27.22 |
> > > |   pi-GAN-light  | 15.8 | 1.447 | 27.35 |
> > > | ShadeGAN (Ours) | 16.3 | 1.001 | 18.68 |
> > >
> > > In fact, it is infeasible to model shading *implicitly* in our setting, because it requires ground truth lighting conditions for supervision, which is not available as we only have raw images. This is also an intuitive explanation that pi-GAN-light and pi-GAN share similar performance as it is hard for pi-GAN-light to interpret the additional input as a light controlling condition rather than another random latent code. In contrast, *explicitly* modeling shading is a kind of prior knowledge that does not require supervision and thus is suitable for our setting.

---

### Decision · Program_Chairs · 2021-09-27

**Decision:**

Accept (Poster)

**Comment:**

The submission initially received mixed reviews. After the rebuttal, all reviewers became positive (though reviewer YNf7 didn't update the score).  The AC agrees with the reviewers that the proposed idea is neat and well-executed. The authors are strongly encouraged to address the remaining concerns in the camera-ready version. This includes adding the results in response to reviewers TVTi, mEVj, and YNf7, addressing the concern on view dependence, among others.